# Probabilistic Logic Neural Networks for Reasoning

**Meng Qu**[1,2], **Jian Tang**[1,3,4]
[1]Mila - Quebec AI Institute [2]University of Montréal
[3]HEC Montréal [4]CIFAR AI Research Chair

## Abstract

Knowledge graph reasoning, which aims at predicting the missing facts through reasoning with the observed facts, is critical to many applications. Such a problem has been widely explored by traditional logic rule-based approaches and recent knowledge graph embedding methods. A principled logic rule-based approach is the Markov Logic Network (MLN), which is able to leverage domain knowledge with first-order logic and meanwhile handle the uncertainty. However, the inference in MLNs is usually very difficult due to the complicated graph structures. Different from MLNs, knowledge graph embedding methods (e.g. TransE, DistMult) learn effective entity and relation embeddings for reasoning, which are much more effective and efficient. However, they are unable to leverage domain knowledge. In this paper, we propose the probabilistic Logic Neural Network (pLogicNet), which combines the advantages of both methods. A pLogicNet defines the joint distribution of all possible triplets by using a Markov logic network with first-order logic, which can be efficiently optimized with the variational EM algorithm. In the E-step, a knowledge graph embedding model is used for inferring the missing triplets, while in the M-step, the weights of logic rules are updated based on both the observed and predicted triplets. Experiments on multiple knowledge graphs prove the effectiveness of pLogicNet over many competitive baselines.

## 1   Introduction

Many real-world entities are interconnected with each other through various types of relationships, forming massive relational data. Naturally, such relational data can be characterized by a set of $(h, r, t)$ triplets, meaning that entity $h$ has relation $r$ with entity $t$. To store the triplets, many knowledge graphs have been constructed such as Freebase [14] and WordNet [24]. These graphs have been proven useful in many tasks, such as question answering [49], relation extraction [34] and recommender systems [4]. However, one big challenge of knowledge graphs is that their coverage is limited. Therefore, one fundamental problem is how to predict the missing links based on the existing triplets.

One type of methods for reasoning on knowledge graphs are the symbolic logic rule-based approaches [12, 17, 35, 41, 46]. These rules can be either handcrafted by domain experts [42] or mined from knowledge graphs themselves [10]. Traditional methods such as expert systems [12, 17] use hard logic rules for prediction. For example, given a logic rule $\forall x, y, \textit{Husband}(x, y) \Rightarrow \textit{Wife}(y, x)$ and a fact that $A$ is the husband of $B$, we can derive that $B$ is the wife of $A$. However, in many cases logic rules can be imperfect or even contradictory, and hence effectively modeling the uncertainty of logic rules is very critical. A more principled method for using logic rules is the Markov Logic Network (MLN) [35, 39], which combines first-order logic and probabilistic graphical models. MLNs learn the weights of logic rules in a probabilistic framework and thus soundly handle the uncertainty. Such methods have been proven effective for reasoning on knowledge graphs. However, the inference process in MLNs is difficult and inefficient due to the complicated graph structure among triplets. Moreover, the results can be unsatisfactory as many missing triplets cannot be inferred by any rules.

Another type of methods for reasoning on knowledge graphs are the recent knowledge graph embedding based methods (e.g., TransE [3], DistMult [48] and ComplEx [44]). These methods learn useful embeddings of entities and relations by projecting existing triplets into low-dimensional spaces.

These embeddings preserve the semantic meanings of entities and relations, and can effectively predict the missing triplets. In addition, they can be efficiently trained with stochastic gradient descent. However, one limitation is that they do not leverage logic rules, which compactly encode domain knowledge and are useful in many applications.

We are seeking an approach that combines the advantages of both worlds, one which is able to exploit first-order logic rules while handling their uncertainty, infer missing triplets effectively, and can be trained in an efficient way. We propose such an approach called the probabilistic Logic Neural Networks (pLogicNet). A pLogicNet defines the joint distribution of a collection of triplets with a Markov Logic Network [35], which associates each logic rule with a weight and can be effectively trained with the variational EM algorithm [26]. In the variational E-step, we infer the plausibility of the unobserved triplets (i.e., hidden variables) with amortized mean-field inference [11, 21, 29], in which the variational distribution is parameterized as a knowledge graph embedding model. In the M-step, we update the weights of logic rules by optimizing the pseudolikelihood [1], which is defined on both the observed triplets and those inferred by the knowledge graph embedding model. The framework can be efficiently trained by stochastic gradient descent. Experiments on four benchmark knowledge graphs prove the effectiveness of pLogicNet over many competitive baselines.

## 2 Related Work

First-order logic rules can compactly encode domain knowledge and have been extensively explored for reasoning. Early methods such as expert systems [12, 17] use hard logic rules for reasoning. However, logic rules can be imperfect or even contradictory. Later studies try to model the uncertainty of logic rules by using Horn clauses [5, 19, 30, 46] or database query languages [31, 41]. A more principled method is the Markov logic network [35, 39], which combines first-order logic with probabilistic graphical models. Despite the effectiveness in a variety of tasks, inference in MLNs remains difficult and inefficient due to the complicated connections between triplets. Moreover, for predicting missing triplets on knowledge graphs, the performance can be limited as many triplets cannot be discovered by any rules. In contrast to them, pLogicNet uses knowledge graph embedding models for inference, which is much more effective by learning useful entity and relation embeddings.

Another category of approach for knowledge graph reasoning is the knowledge graph embedding method [3, 8, 28, 40, 44, 45, 48], which aims at learning effective embeddings of entities and relations. Generally, these methods design different scoring functions to model different relation patterns for reasoning. For example, TransE [3] defines each relation as a translation vector, which can effectively model the composition and inverse relation patterns. DistMult [48] models the symmetric relation with a bilinear scoring function. ComplEx [44] models the asymmetric relations by using a bilinear scoring function in complex space. RotatE [40] further models multiple relation patterns by defining each relation as a rotation in complex spaces. Despite the effectiveness and efficiency, these methods are not able to leverage logic rules, which are beneficial in many tasks. Recently, there are a few studies on combining logic rules and knowledge graph embedding [9, 15]. However, they cannot effectively handle the uncertainty of logic rules. Compared with them, pLogicNet is able to use logic rules and also handle their uncertainty in a more principled way through Markov logic networks.

Some recent work also studies using reinforcement learning for reasoning on knowledge graphs [6, 23, 38, 47], where an agent is trained to search for reasoning paths. However, the performance of these methods is not so competitive. Our pLogicNets are easier to train and also more effective.

Lastly, there are also some recent studies trying to combine statistical relational learning and graph neural networks for semi-supervised node classification [33], or using Markov networks for visual dialog reasoning [32, 51]. Our work shares similar idea with these studies, but we focus on a different problem, i.e., reasoning with first-order logic on knowledge graphs. There is also a concurrent work using graph neural networks for logic reasoning [50]. Compared to this study which emphasizes more on the inference problem, our work focuses on both the inference and the learning problems.

## 3 Preliminary

### 3.1 Problem Definition

A knowledge graph is a collection of relational facts, each of which is represented as a triplet $(h, r, t)$. Due to the high cost of knowledge graph construction, the coverage of knowledge graphs is usually limited. Therefore, a critical problem on knowledge graphs is to predict the missing facts.

Formally, given a knowledge graph $(E, R, O)$, where $E$ is a set of entities, $R$ is a set of relations, and $O$ is a set of observed $(h, r, t)$ triplets, the goal is to infer the missing triplets by reasoning with the observed triplets. Following existing studies [27], the problem can be reformulated in a probabilistic way. Each triplet $(h, r, t)$ is associated with a binary indicator variable $\mathbf{v}_{(h,r,t)}$. $\mathbf{v}_{(h,r,t)} = 1$ means $(h, r, t)$ is true, and $\mathbf{v}_{(h,r,t)} = 0$ otherwise. Given some true facts $\mathbf{v}_O = \{\mathbf{v}_{(h,r,t)} = 1\}_{(h,r,t) \in O}$, we aim to predict the labels of the remaining hidden triplets $H$, i.e., $\mathbf{v}_H = \{\mathbf{v}_{(h,r,t)}\}_{(h,r,t) \in H}$. We will discuss how to generate the hidden triplets $H$ later in Sec. 4.4.

This problem has been extensively studied in both traditional logic rule-based methods and recent knowledge graph embedding methods. For logic rule-based methods, we mainly focus on one representative approach, the Markov logic network [35]. Essentially, both types of methods aim to model the joint distribution of the observed and hidden triplets $p(\mathbf{v}_O, \mathbf{v}_H)$. Next, we briefly introduce the Markov logic network (MLN) [35] and the knowledge graph embedding methods [3, 40, 48].

### 3.2 Markov Logic Network

In the MLN, a Markov network is designed to define the joint distribution of the observed and the hidden triplets, where the potential function is defined by the first-order logic. Some common logic rules to encode domain knowledge include: (1) **Composition Rules**. A relation $r_k$ is a composition of $r_i$ and $r_j$ means that for any three entities $x, y, z$, if $x$ has relation $r_i$ with $y$, and $y$ has relation $r_j$ with $z$, then $x$ has relation $r_k$ with $z$. Formally, we have $\forall x, y, z \in E, \mathbf{v}_{(x,r_i,y)} \wedge \mathbf{v}_{(y,r_j,z)} \Rightarrow \mathbf{v}_{(x,r_k,z)}$. (2) **Inverse Rules**. A relation $r_j$ is an inverse of $r_i$ indicates that for two entities $x, y$, if $x$ has relation $r_i$ with $y$, then $y$ has relation $r_j$ with $x$. We can represent the rule as $\forall x, y \in E, \mathbf{v}_{(x,r_i,y)} \Rightarrow \mathbf{v}_{(y,r_j,x)}$. (3) **Symmetric Rules**. A relation $r$ is symmetric means that for any entity pair $x, y$, if $x$ has relation $r$ with $y$, then $y$ also has relation $r$ with $x$. Formally, we have $\forall x, y \in E, \mathbf{v}_{(x,r,y)} \Rightarrow \mathbf{v}_{(y,r,x)}$. (4) **Subrelation Rules**. A relation $r_j$ is a subrelation of $r_i$ indicates that for any entity pair $x, y$, if $x$ and $y$ have relation $r_i$, then they also have relation $r_j$. Formally, we have $\forall x, y \in E, \mathbf{v}_{(x,r_i,y)} \Rightarrow \mathbf{v}_{(x,r_j,y)}$.

For each logic rule $l$, we can obtain a set of possible groundings $G_l$ by instantiating the entity placeholders in the logic rule with real entities in knowledge graphs. For example, for a subre-lation rule, $\forall x, y, \mathbf{v}_{(x,\text{Born in},y)} \Rightarrow \mathbf{v}_{(x,\text{Live in},y)}$, two groundings in $G_l$ can be $\mathbf{v}_{(\text{Newton,Born in,UK})} \Rightarrow \mathbf{v}_{(\text{Newton,Live in,UK})}$ and $\mathbf{v}_{(\text{Einstein,Born in,German})} \Rightarrow \mathbf{v}_{(\text{Einstein,Live in,German})}$. We see that the former one is true while the latter one is false. To handle such uncertainty of logic rules, Markov logic networks introduce a weight $w_l$ for each rule $l$, and then the joint distribution of all triplets is defined as follows:

$$p(\mathbf{v}_O, \mathbf{v}_H) = \frac{1}{Z} \exp \left( \sum_{l \in L} w_l \sum_{g \in G_l} \mathbb{1}\{g \text{ is true}\} \right) = \frac{1}{Z} \exp \left( \sum_{l \in L} w_l n_l(\mathbf{v}_O, \mathbf{v}_H) \right), \quad (1)$$

where $n_l$ is the number of true groundings of the logic rule $l$ based on the values of $\mathbf{v}_O$ and $\mathbf{v}_H$.

With such a formulation, predicting the missing triplets essentially becomes inferring the posterior distribution $p(\mathbf{v}_H | \mathbf{v}_O)$. Exact inference is usually infeasible due to the complicated graph structures, and hence approximation inference is often used such as MCMC [13] and loopy belief propagation [25].

### 3.3 Knowledge Graph Embedding

Different from the logic rule-based approaches, the knowledge graph embedding methods learn embeddings of entities and relations with the observed facts $\mathbf{v}_O$, and then predict the missing facts with the learned entity and relation embeddings. Formally, each entity $e \in E$ and relation $r \in R$ is associated with an embedding $\mathbf{x}_e$ and $\mathbf{x}_r$. Then the joint distribution of all the triplets is defined as:

$$p(\mathbf{v}_O, \mathbf{v}_H) = \prod_{(h,r,t) \in O \cup H} \text{Ber}(\mathbf{v}_{(h,r,t)} | f(\mathbf{x}_h, \mathbf{x}_r, \mathbf{x}_t)), \quad (2)$$

where Ber stands for the Bernoulli distribution, $f(\mathbf{x}_h, \mathbf{x}_r, \mathbf{x}_t)$ computes the probability that the triplet $(h, r, t)$ is true, with $f(\cdot, \cdot, \cdot)$ being a scoring function on the entity and relation embeddings. For example in TransE, the score function $f$ can be formulated as $\sigma(\gamma - ||\mathbf{x}_h + \mathbf{x}_r - \mathbf{x}_t||)$ according to [40], where $\sigma$ is the sigmoid function and $\gamma$ is a fixed bias. To learn the entity and relation embeddings, these methods typically treat observed triplets as positive examples and the hidden triplets as negative ones. In other words, these methods seek to maximize $\log p(\mathbf{v}_O = 1, \mathbf{v}_H = 0)$. The whole framework can be efficiently optimized with the stochastic gradient descent algorithm.

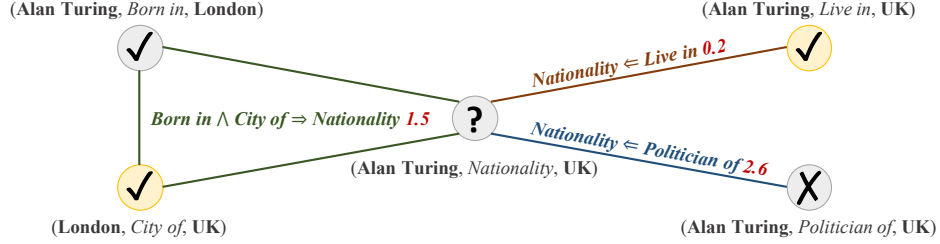

Figure 1: Framework overview. Each possible triplet is associated with a binary indicator (circles), indicating whether it is true (✔) or not (✘). The observed (yellow circles) and hidden (grey circles) indicators are connected by a set of logic rules, with each rule having a weight (red number). For the center triplet, the KGE model predicts its indicator through embeddings, while the logic rules consider the Markov blanket of the triplet (all connected triplets). If any indicator in the Markov blanket is hidden, we simply fill it with the prediction from the KGE model. In the **E-step**, we use the logic rules to predict the center indicator, and treat it as extra training data for the KGE model. In the **M-step**, we annotate all hidden indicators with the KGE model, and then update the weights of rules.

## 4 Model

In this section, we introduce our proposed approach pLogicNet for knowledge graph reasoning, which combines the logic rule-based methods and the knowledge graph embedding methods. To leverage the domain knowledge provided by first-order logic rules, pLogicNet formulates the joint distribution of all triplets with a Markov logic network [35], which is trained with the variational EM algorithm [26], alternating between a variational E-step and an M-step. In the varational E-step, we employ a knowledge graph embedding model to infer the missing triplets, during which the knowledge preserved by the logic rules can be effectively distilled into the learned embeddings. In the M-step, the weights of the logic rules are updated based on both the observed triplets and those inferred by the knowledge graph embedding model. In this way, the knowledge graph embedding model provides extra supervision for weight learning. An overview of pLogicNet is given in Fig. 1.

### 4.1 Variational EM

Given a set of first-order logic rules $L = \{l_i\}_{i=1}^{|L|}$, our approach uses a Markov logic network [35] as in Eq. (1) to model the joint distribution of both the observed and hidden triplets:

$$p_w(\mathbf{v}_O, \mathbf{v}_H) = \frac{1}{Z} \exp\left(\sum_l w_l n_l(\mathbf{v}_O, \mathbf{v}_H)\right), \tag{3}$$

where $w_l$ is the weight of rule $l$. The model can be trained by maximizing the log-likelihood of the observed indicator variables, i.e., $\log p_w(\mathbf{v}_O)$. However, directly optimizing the objective is infeasible, as we need to integrate over all the hidden indicator variables $\mathbf{v}_H$. Therefore, we instead optimize the evidence lower bound (ELBO) of the log-likelihood function, which is given as follows:

$$\log p_w(\mathbf{v}_O) \geq \mathcal{L}(q_\theta, p_w) = \mathbb{E}_{q_\theta(\mathbf{v}_H)}[\log p_w(\mathbf{v}_O, \mathbf{v}_H) - \log q_\theta(\mathbf{v}_H)], \tag{4}$$

where $q_\theta(\mathbf{v}_H)$ is a variational distribution of the hidden variables $\mathbf{v}_H$. The equation holds when $q_\theta(\mathbf{v}_H)$ equals to the true posterior distribution $p_w(\mathbf{v}_H|\mathbf{v}_O)$. Such a lower bound can be effectively optimized with the variational EM algorithm [26], which consists of a variational E-step and an M-step. In the variational E-step, which is known as the inference procedure, we fix $p_w$ and update $q_\theta$ to minimize the KL divergence between $q_\theta(\mathbf{v}_H)$ and $p_w(\mathbf{v}_H|\mathbf{v}_O)$. In the M-step, which is known as the learning procedure, we fix $q_\theta$ and update $p_w$ to maximize the log-likelihood function of all the triplets, i.e., $\mathbb{E}_{q_\theta(\mathbf{v}_H)}[\log p_w(\mathbf{v}_O, \mathbf{v}_H)]$. Next, we introduce the details of both steps.

### 4.2 E-step: Inference Procedure

For inference, we aim to infer the posterior distribution of the hidden variables, i.e., $p_w(\mathbf{v}_H|\mathbf{v}_O)$. As exact inference is intractable, we approximate the true posterior distribution with a mean-field [29]

variational distribution $q_\theta(\mathbf{v}_H)$, in which each $\mathbf{v}_{(h,r,t)}$ is inferred independently for $(h,r,t) \in H$. To further improve inference, we use amortized inference [11, 21], and parameterize $q_\theta(\mathbf{v}_{(h,r,t)})$ with a knowledge graph embedding model. Formally, $q_\theta(\mathbf{v}_H)$ is formulated as below:

$$q_\theta(\mathbf{v}_H) = \prod_{(h,r,t)\in H} q_\theta(\mathbf{v}_{(h,r,t)}) = \prod_{(h,r,t)\in H} \mathrm{Ber}(\mathbf{v}_{(h,r,t)}|f(\mathbf{x}_h, \mathbf{x}_r, \mathbf{x}_t)), \qquad (5)$$

where Ber stands for the Bernoulli distribution, and $f(\cdot,\cdot,\cdot)$ is a scoring function defined on triplets as introduced in Sec. 3.3. By minimizing the KL divergence between the variational distribution $q_\theta(\mathbf{v}_H)$ and the true posterior $p_w(\mathbf{v}_H|\mathbf{v}_O)$, the optimal $q_\theta(\mathbf{v}_H)$ is given by the fixed-point condition:

$$\log q_\theta(\mathbf{v}_{(h,r,t)}) = \mathbb{E}_{q_\theta(\mathbf{v}_{\mathrm{MB}(h,r,t)})}[\log p_w(\mathbf{v}_{(h,r,t)}|\mathbf{v}_{\mathrm{MB}(h,r,t)})] + \mathrm{const} \quad \text{for all } (h,r,t) \in H, \quad (6)$$

where $\mathrm{MB}(h,r,t)$ is the Markov blanket of $(h,r,t)$, which contains the triplets that appear together with $(h,r,t)$ in any grounding of the logic rules. For example, from a grounding $\mathbf{v}_{(\mathrm{Newton,Born\ in,UK})} \Rightarrow \mathbf{v}_{(\mathrm{Newton,Live\ in,UK})}$, we can know both triplets are in the Markov blanket of each other.

With Eq. (6), our goal becomes finding a distribution $q_\theta$ that satisfies the condition. However, Eq. (6) involves the expectation with respect to $q_\theta(\mathbf{v}_{\mathrm{MB}(h,r,t)})$. To simplify the condition, we follow [16] and estimate the expectation with a sample $\hat{\mathbf{v}}_{\mathrm{MB}(h,r,t)} = \{\hat{\mathbf{v}}_{(h',r',t')}\}_{(h',r',t')\in\mathrm{MB}(h,r,t)}$. Specifically, for each $(h',r',t') \in \mathrm{MB}(h,r,t)$, if it is observed, we set $\hat{\mathbf{v}}_{(h',r',t')} = 1$, and otherwise $\hat{\mathbf{v}}_{(h',r',t')} \sim q_\theta(\mathbf{v}_{(h',r',t')})$. In this way, the right side of Eq. (6) is approximated as $\log p_w(\mathbf{v}_{(h,r,t)}|\hat{\mathbf{v}}_{\mathrm{MB}(h,r,t)})$, and thus the optimality condition can be further simplified as $q_\theta(\mathbf{v}_{(h,r,t)}) \approx p_w(\mathbf{v}_{(h,r,t)}|\hat{\mathbf{v}}_{\mathrm{MB}(h,r,t)})$.

Intuitively, for each hidden triplet $(h,r,t)$, the knowledge graph embedding model predicts $\mathbf{v}_{(h,r,t)}$ through the entity and relation embeddings (i.e., $q_\theta(\mathbf{v}_{(h,r,t)})$), while the logic rules make the prediction by utilizing the triplets connected with $(h,r,t)$ (i.e., $p_w(\mathbf{v}_{(h,r,t)}|\hat{\mathbf{v}}_{\mathrm{MB}(h,r,t)})$). If any triplet $(h',r',t')$ connected with $(h,r,t)$ is unobserved, we simply fill in $\mathbf{v}_{(h',r',t')}$ with a sample $\hat{\mathbf{v}}_{(h',r',t')} \sim q_\theta(\mathbf{v}_{(h',r',t')})$. Then, the simplified optimality condition tells us that for the optimal knowledge graph embedding model, it should reach a consensus with the logic rules on the distribution of $\mathbf{v}_{(h,r,t)}$ for every $(h,r,t)$, i.e., $q_\theta(\mathbf{v}_{(h,r,t)}) \approx p_w(\mathbf{v}_{(h,r,t)}|\hat{\mathbf{v}}_{\mathrm{MB}(h,r,t)})$.

To learn the optimal $q_\theta$, we use a method similar to [36]. We start by computing $p_w(\mathbf{v}_{(h,r,t)}|\hat{\mathbf{v}}_{\mathrm{MB}(h,r,t)})$ with the current $q_\theta$. Then, we fix the value as target, and update $q_\theta$ to minimize the reverse KL divergence of $q_\theta(\mathbf{v}_{(h,r,t)})$ and the target $p_w(\mathbf{v}_{(h,r,t)}|\hat{\mathbf{v}}_{\mathrm{MB}(h,r,t)})$, leading to the following objective:

$$O_{\theta,U} = \sum_{(h,r,t)\in H} \mathbb{E}_{p_w(\mathbf{v}_{(h,r,t)}|\hat{\mathbf{v}}_{\mathrm{MB}(h,r,t)})}[\log q_\theta(\mathbf{v}_{(h,r,t)})]. \qquad (7)$$

To optimize this objective, we first compute $p_w(\mathbf{v}_{(h,r,t)}|\hat{\mathbf{v}}_{\mathrm{MB}(h,r,t)})$ for each hidden triplet $(h,r,t)$. If $p_w(\mathbf{v}_{(h,r,t)} = 1|\hat{\mathbf{v}}_{\mathrm{MB}(h,r,t)}) \geq \tau_{\mathrm{triplet}}$ with $\tau_{\mathrm{triplet}}$ being a hyperparameter, then we treat $(h,r,t)$ as a positive example and train the knowledge graph embedding model to maximize the log-likelihood $\log q_\theta(\mathbf{v}_{(h,r,t)} = 1)$. Otherwise the triplet is treated as a negative example. In this way, the knowledge captured by logic rules can be effectively distilled into the knowledge graph embedding model.

We can also use the observed triplets in $O$ as positive examples to enhance the knowledge graph embedding model. Therefore, we also optimize the following objective function:

$$O_{\theta,L} = \sum_{(h,r,t)\in O} \log q_\theta(\mathbf{v}_{(h,r,t)} = 1). \qquad (8)$$

By adding Eq. (7) and (8), we obtain the overall objective function for $q_\theta$, i.e., $O_\theta = O_{\theta,U} + O_{\theta,L}$.

### 4.3 M-step: Learning Procedure

In the learning procedure, we will fix $q_\theta$, and update the weights of logic rules $w$ by maximizing the log-likelihood function, i.e., $\mathbb{E}_{q_\theta(\mathbf{v}_H)}[\log p_w(\mathbf{v}_O, \mathbf{v}_H)]$. However, directly optimizing the log-likelihood function can be difficult, as we need to deal with the partition function, i.e., $Z$ in Eq. (3). Therefore, we follow existing studies [22, 35] and instead optimize the pseudolikelihood function [1]:

$$\ell_{PL}(w) \triangleq \mathbb{E}_{q_\theta(\mathbf{v}_H)}[\sum_{h,r,t} \log p_w(\mathbf{v}_{(h,r,t)}|\mathbf{v}_{O\cup H\setminus(h,r,t)})] = \mathbb{E}_{q_\theta(\mathbf{v}_H)}[\sum_{h,r,t} \log p_w(\mathbf{v}_{(h,r,t)}|\mathbf{v}_{\mathrm{MB}(h,r,t)})],$$

where the second equation is derived from the independence property of the MLN in the Eq. (3).

We optimize $w$ through the gradient descent algorithm. For each expected conditional distribution $\mathbb{E}_{q_\theta(\mathbf{v}_H)}[\log p_w(\mathbf{v}_{(h,r,t)}|\mathbf{v}_{\text{MB}(h,r,t)})]$, suppose $\mathbf{v}_{(h,r,t)}$ connects with $\mathbf{v}_{\text{MB}(h,r,t)}$ through a set of rules. For each of such rules $l$, the derivative with respect to $w_l$ is computed as:

$$\nabla_{w_l}\mathbb{E}_{q_\theta(\mathbf{v}_H)}[\log p_w(\mathbf{v}_{(h,r,t)}|\mathbf{v}_{\text{MB}(h,r,t)})] \simeq y_{(h,r,t)} - p_w(\mathbf{v}_{(h,r,t)} = 1|\hat{\mathbf{v}}_{\text{MB}(h,r,t)}) \qquad (9)$$

where $y_{(h,r,t)} = 1$ if $(h,r,t)$ is an observed triplet and $y_{(h,r,t)} = q_\theta(\mathbf{v}_{(h,r,t)} = 1)$ if $(h,r,t)$ is a hidden one. $\hat{\mathbf{v}}_{\text{MB}(h,r,t)} = \{\hat{\mathbf{v}}_{(h',r',t')}\}_{(h',r',t')\in\text{MB}(h,r,t)}$ is a sample from $q_\theta$. For each $(h',r',t') \in \text{MB}(h,r,t)$, $\hat{\mathbf{v}}_{(h',r',t')} = 1$ if $(h',r',t')$ is observed, and otherwise $\hat{\mathbf{v}}_{(h',r',t')} \sim q_\theta(\mathbf{v}_{(h',r',t')})$.

Intuitively, for each observed triplet $(h,r,t) \in O$, we seek to maximize $p_w(\mathbf{v}_{(h,r,t)} = 1|\hat{\mathbf{v}}_{\text{MB}(h,r,t)})$. For each hidden triplet $(h,r,t) \in H$, we treat $q_\theta(\mathbf{v}_{(h,r,t)} = 1)$ as target for updating the probability $p_w(\mathbf{v}_{(h,r,t)} = 1|\hat{\mathbf{v}}_{\text{MB}(h,r,t)})$. In this way, the knowledge graph embedding model $q_\theta$ essentially provides extra supervision to benefit learning the weights of logic rules.

## 4.4 Optimization and Prediction

During training, we iteratively perform the E-step and the M-step until convergence. Note that there are a huge number of possible hidden triplets (i.e., $|E| \times |R| \times |E| - |O|$), and handling all of them is impractical for optimization. Therefore, we only include a small number of triplets in the hidden set $H$. Specifically, an unobserved triplet $(h,r,t)$ is added to $H$ if we can find a grounding $[premise] \Rightarrow [hypothesis]$, where the hypothesis is $(h,r,t)$ and the premise only contains triplets in the observed set $O$. In practice, we can construct $H$ with brute-force search as in [15].

After training, according to the fixed-point condition given in Eq. (6), the posterior distribution $p_w(\mathbf{v}_{(h,r,t)}|\mathbf{v}_O)$ for $(h,r,t) \in H$ can be characterized by either $q_\theta(\mathbf{v}_{(h,r,t)})$ or $p_w(\mathbf{v}_{(h,r,t)}|\hat{\mathbf{v}}_{\text{MB}(h,r,t)})$ with $\hat{\mathbf{v}}_{\text{MB}(h,r,t)} \sim q_\theta(\mathbf{v}_{\text{MB}(h,r,t)})$. Although we try to encourage the consensus of $p_w$ and $q_\theta$ during training, they may still give different predictions as different information is used. Therefore, we use both of them for prediction, and we approximate the true posterior distribution $p_w(\mathbf{v}_{(h,r,t)}|\mathbf{v}_O)$ as:

$$p_w(\mathbf{v}_{(h,r,t)}|\mathbf{v}_O) \propto \left\{ q_\theta(\mathbf{v}_{(h,r,t)}) + \lambda p_w(\mathbf{v}_{(h,r,t)}|\hat{\mathbf{v}}_{\text{MB}(h,r,t)}) \right\}, \qquad (10)$$

where $\lambda$ is a hyperparameter controlling the relative weight of $q_\theta(\mathbf{v}_{(h,r,t)})$ and $p_w(\mathbf{v}_{(h,r,t)}|\hat{\mathbf{v}}_{\text{MB}(h,r,t)})$. In practice, we also expect to infer the plausibility of the triplets outside $H$. For each of such triplets $(h,r,t)$, we can still compute $q_\theta(\mathbf{v}_{(h,r,t)})$ through the learned embeddings, but we cannot make predictions with the logic rules, so we simply replace $p_w(\mathbf{v}_{(h,r,t)} = 1|\hat{\mathbf{v}}_{\text{MB}(h,r,t)})$ with 0.5 in Eq. 10.

# 5 Experiment

## 5.1 Experiment Settings

**Datasets.** In experiments, we evaluate the pLogicNet on four benchmark datasets. The FB15k [3] and FB15k-237 [43] datasets are constructed from Freebase [2]. WN18 [3] and WN18RR [8] are constructed from WordNet [24]. The detailed statistics of the datasets are summarized in appendix.

**Evaluation Metrics.** We compare different methods on the task of knowledge graph reasoning. For each test triplet, we mask the head or the tail entity, and let each compared method predict the masked entity. Following existing studies [3, 48], we use the filtered setting during evaluation. The Mean Rank (MR), Mean Reciprocal Rank (MRR) and Hit@K (H@K) are treated as the evaluation metrics.

**Compared Algorithms.** We compare with both the knowledge graph embedding methods and rule-based methods. For the knowledge graph embedding methods, we choose five representative methods to compare with, including TransE [3], DistMult [48], HolE [28], ComplEx [44] and ConvE [8]. For the rule-based methods, we compare with the Markov logic network (MLN) [35] and the Bayesian logic programming (BLP) method [7], which model logic rules with Markov networks and Bayesian networks respectively. Besides, we also compare with RUGE [15] and NNE-AER [9], which are hybrid methods that combine knowledge graph embedding and logic rules. As only the results on the FB15k dataset are reported in the RUGE paper, we only compare with RUGE on that dataset. For our approach, we consider two variants, where **pLogicNet** uses only $q_\theta$ to infer the plausibility of unobserved triplets during evaluation, while **pLogicNet**$^*$ uses both $q_\theta$ and $p_w$ through Eq. (10).

**Experimental Setup of pLogicNet.** To generate the candidate rules in the pLogicNet, we search for all the possible composition rules, inverse rules, symmetric rules and subrelations rules from the observed triplets, which is similar to [10, 15]. Then, we compute the empirical precision of each rule, i.e. $p_l = \frac{|S_l \cap O|}{|S_l|}$, where $S_l$ is the set of triplets extracted by the rule $l$ and $O$ is the set of the observed triplets. We only keep rules whose empirical precision is larger than a threshold $\tau_{\text{rule}}$. TransE [3] is used as the default knowledge graph embedding model to parameterize $q_\theta$. We update the weights of logic rules with gradient descent. The detailed hyperparameters settings are available in the appendix.

## 5.2 Results

### 5.2.1 Comparing pLogicNet with Other Methods

Table 1: Results of reasoning on the FB15k and WN18 datasets. The results of the KGE and the Hybrid methods except for TransE are directly taken from the corresponding papers. H@K is in %.

| Category | Algorithm | FB15k | | | | | WN18 | | | | |
|---|---|---|---|---|---|---|---|---|---|---|---|
| | | MR | MRR | H@1 | H@3 | H@10 | MR | MRR | H@1 | H@3 | H@10 |
| KGE | TransE [3] | 40 | 0.730 | 64.5 | 79.3 | 86.4 | 272 | 0.772 | 70.1 | 80.8 | 92.0 |
| | DistMult [18] | 42 | 0.798 | - | - | 89.3 | 655 | 0.797 | - | - | 94.6 |
| | HolE [28] | - | 0.524 | 40.2 | 61.3 | 73.9 | - | 0.938 | 93.0 | 94.5 | 94.9 |
| | ComplEx [44] | - | 0.692 | 59.9 | 75.9 | 84.0 | - | 0.941 | 93.6 | 94.5 | 94.7 |
| | ConvE [8] | 51 | 0.657 | 55.8 | 72.3 | 83.1 | 374 | 0.943 | 93.5 | 94.6 | 95.6 |
| Rule-based | BLP [7] | 415 | 0.242 | 15.1 | 26.9 | 42.4 | 736 | 0.643 | 53.7 | 71.7 | 83.0 |
| | MLN [35] | 352 | 0.321 | 21.0 | 37.0 | 55.0 | 717 | 0.657 | 55.4 | 73.1 | 83.9 |
| Hybrid | RUGE [15] | - | 0.768 | 70.3 | 81.5 | 86.5 | - | - | - | - | - |
| | NNE-AER [9] | - | 0.803 | 76.1 | 83.1 | 87.4 | - | 0.943 | **94.0** | 94.5 | 94.8 |
| Ours | pLogicNet | **33** | 0.792 | 71.4 | 85.7 | 90.1 | 255 | 0.832 | 71.6 | 94.4 | 95.7 |
| | pLogicNet* | **33** | **0.844** | **81.2** | **86.2** | **90.2** | 254 | **0.945** | 93.9 | **94.7** | **95.8** |

Table 2: Results of reasoning on the FB15k-237 and WN18RR datasets. The results of the KGE methods except for TransE are directly taken from the corresponding papers. H@K is in %.

| Category | Algorithm | FB15k-237 | | | | | WN18RR | | | | |
|---|---|---|---|---|---|---|---|---|---|---|---|
| | | MR | MRR | H@1 | H@3 | H@10 | MR | MRR | H@1 | H@3 | H@10 |
| KGE | TransE [3] | 181 | 0.326 | 22.9 | 36.3 | 52.1 | 3410 | 0.223 | 1.3 | 40.1 | 53.1 |
| | DistMult [18] | 254 | 0.241 | 15.5 | 26.3 | 41.9 | 5110 | 0.43 | 39 | 44 | 49 |
| | ComplEx [44] | 339 | 0.247 | 15.8 | 27.5 | 42.8 | 5261 | **0.44** | **41** | **46** | 51 |
| | ConvE [8] | 244 | 0.325 | **23.7** | 35.6 | 50.1 | 4187 | 0.43 | 40 | 44 | 52 |
| Rule-based | BLP [7] | 1985 | 0.092 | 6.2 | 9.8 | 15.0 | 12051 | 0.254 | 18.7 | 31.3 | 35.8 |
| | MLN [35] | 1980 | 0.098 | 6.7 | 10.3 | 16.0 | 11549 | 0.259 | 19.1 | 32.2 | 36.1 |
| Ours | pLogicNet | **173** | 0.330 | 23.1 | **36.9** | **52.8** | 3436 | 0.230 | 1.5 | 41.1 | 53.1 |
| | pLogicNet* | **173** | **0.332** | **23.7** | 36.7 | 52.4 | **3408** | **0.441** | 39.8 | 44.6 | **53.7** |

The main results on the four datasets are presented in Tab. 1 and 2. We can see that the pLogicNet significantly outperforms the rule-based methods, as pLogicNet uses a knowledge graph embedding model to improve inference. pLogicNet also outperforms all the knowledge graph embedding methods in most cases, where the improvement comes from the capability of exploring the knowledge captured by the logic rules. Moreover, our approach is superior to both hybrid methods (RUGE and NNE-AER) under most metrics, as it handles the uncertainty of logic rules in a more principled way.

Comparing pLogicNet and pLogicNet*, pLogicNet* uses both $q_\theta$ and $p_w$ to predict the plausibility of hidden triplets, which outperforms pLogicNet in most cases. The reason is that the information captured by $q_\theta$ and $p_w$ is different and complementary, so combining them yields better performance.

### 5.2.2 Analysis of Different Rule Patterns

Table 3: Analysis of different rule patterns. H@K is in %.

| Rule Pattern | FB15k | | | | | FB15k-237 | | | | |
|---|---|---|---|---|---|---|---|---|---|---|
| | MR | MRR | H@1 | H@3 | H@10 | MR | MRR | H@1 | H@3 | H@10 |
| Without | 40 | 0.730 | 64.7 | 79.4 | 86.4 | 181 | 0.326 | 22.9 | 36.3 | 52.1 |
| Composition | 40 | 0.752 | 69.3 | 78.7 | 86.0 | 173 | **0.335** | **24.1** | **37.1** | **52.5** |
| Inverse | **39** | **0.813** | **77.7** | **83.1** | **88.1** | 175 | 0.332 | 23.8 | 36.7 | 52.4 |
| Symmetric | 40 | 0.793 | 75.0 | 81.7 | 87.1 | 175 | 0.333 | 23.8 | 36.8 | 52.4 |
| Subrelation | 40 | 0.761 | 70.2 | 79.8 | 86.6 | **172** | 0.334 | 23.9 | 36.8 | **52.5** |

In pLogicNet, four types of rule patterns are used. Next, we systematically study the effect of each rule pattern. We take the FB15k and FB15k-237 datasets as examples, and report the results obtained with each single rule pattern in Tab. 3. On both datasets, most rule patterns can lead to significant improvement compared to the model without logic rules. Moreover, the effects of different rule patterns are quite different across datasets. On FB15k, the inverse and symmetric rules are more important, whereas on FB15k-237, the composition and subrelation rules are more effective.

### 5.2.3 Inference with Different Knowledge Graph Embedding Methods

Table 4: Comparison of using different knowledge graph embedding methods. H@K is in %.

| KGE Method | Algorithm | FB15k | | | | | WN18RR | | | | |
|---|---|---|---|---|---|---|---|---|---|---|---|
| | | MR | MRR | H@1 | H@3 | H@10 | MR | MRR | H@1 | H@3 | H@10 |
| TransE | pLogicNet | 33 | 0.792 | 71.4 | 85.7 | 90.1 | 3436 | 0.230 | 1.5 | 41.1 | 53.1 |
| | pLogicNet* | 33 | 0.844 | 81.2 | 86.2 | 90.2 | 3408 | 0.441 | 39.8 | 44.6 | 53.7 |
| DistMult | pLogicNet | 40 | 0.791 | 73.1 | 83.2 | 89.5 | 4902 | 0.442 | 39.8 | 45.5 | 53.5 |
| | pLogicNet* | 39 | 0.815 | 76.8 | 84.6 | 89.8 | 4894 | 0.443 | 39.9 | 45.5 | 53.6 |
| ComplEx | pLogicNet | 39 | 0.776 | 70.6 | 81.7 | 88.5 | 5266 | 0.471 | 43.0 | 49.2 | 55.7 |
| | pLogicNet* | 45 | 0.788 | 73.5 | 82.1 | 88.5 | 5233 | 0.475 | 43.5 | 49.2 | 55.7 |

In this part, we compare the performance of pLogicNet with different knowledge graph embedding methods for inference. We use TransE as the default model and compare with two other widely-used knowledge graph embedding methods, DistMult [48] and ComplEx [44]. The results on the FB15k and WN18RR datasets are presented in Tab. 4. Comparing with the results in Tab. 1 and 2, we see that pLogicNet improves the performance of all the three methods by using logic rules. Moreover, the pLogicNet achieves very robust performance with any of the three methods for inference.

| Iteration | FB15k | | WN18 | |
|---|---|---|---|---|
| | # Triplets | Precision | # Triplets | Precision |
| 1 | 64,929 | 79.21% | 11,146 | 80.99% |
| 2 | 74,717 | 79.31% | 11,430 | 82.06% |
| 3 | 76,268 | 79.10% | 11,432 | 82.09% |

Table 5: Effect of KGE on logic rules.

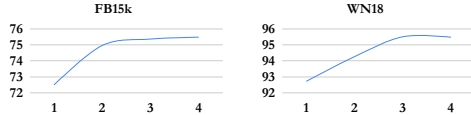

Figure 2: Convergence analysis.

### 5.2.4 Effect of Knowledge Graph Embedding on Logic Rules

In the M-step of pLogicNet, we use the learned embeddings to annotate the hidden triplets, and further update the weights of logic rules. Next, we analyze the effect of knowledge graph embeddings on logic rules. Recall that in the E-step, the logic rules are used to annotate the hidden triplets through Eq. (7), and thus collect extra positive training data for embedding learning. To evaluate the performance of logic rules, in each iteration we report the number of positive triplets discovered by logic rules, as well as the precision of the triplets in Tab. 5. We see that as training proceeds, the logic rules can find more triplets with stable precision. This observation proves that the knowledge graph embedding model can indeed provide effective supervision for learning the weights of logic rules.

### 5.2.5 Convergence Analysis

Finally, we present the convergence curves of pLogicNet* on the FB15k and WN18 datasets in Fig. 2. The horizontal axis represents the iteration, and the vertical axis shows the value of Hit@1 (in %). We see that on both datasets, our approach takes only 2-3 iterations to converge, which is very efficient.

## 6 Conclusion

This paper studies knowledge graph reasoning, and an approach called the pLogicNet is proposed to integrate existing rule-based methods and knowledge graph embedding methods. pLogicNet models the distribution of all the possible triplets with a Markov logic network, which is efficiently optimized with the variational EM algorithm. In the E-step, a knowledge graph embedding model is used to infer the hidden triplets, whereas in the M-step, the weights of rules are updated based on the observed and inferred triplets. Experimental results prove the effectiveness of pLogicNet. In the future, we plan to explore more advanced models for inference, such as relational GCN [37, 50] and RotatE [40].

## Acknowledgements

We would like to thank all the reviewers for the insightful comments. We also thank Prof. Guillaume Rabusseau and Weiping Song for their valuable feedback. Jian Tang is supported by the Natural Sciences and Engineering Research Council of Canada, and the Canada CIFAR AI Chair Program.

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
