[Supplementary Material]

# 1 Appendix

## 1.1 Proof of the Optimality Condition for $q_\theta$

In pLogicNet, we use a mean-field variational distribution $q_\theta(\mathbf{v}_H)$ to approximate the true posterior distribution $p_w(\mathbf{v}_H|\mathbf{v}_O)$. In this section, we prove that the optimal variational distribution is given by the following fixed-point condition:

$$q_\theta(\mathbf{v}_H) = \prod_{(h,r,t)\in H} q_\theta(\mathbf{v}_{(h,r,t)}) = \prod_{(h,r,t)\in H} \mathrm{Ber}(\mathbf{v}_{(h,r,t)}|f(\mathbf{x}_h,\mathbf{x}_r,\mathbf{x}_t)), \tag{1}$$

To prove the claim, recall that our goal for $q_\theta$ is to minimize the KL divergence between $q_\theta(\mathbf{v}_H)$ and $p_w(\mathbf{v}_H|\mathbf{v}_O)$. If we consider each individual indicator variable $\mathbf{v}_{(h,r,t)}$, the objective function for $q_\theta(\mathbf{v}_{(h,r,t)})$ is given as follows:

$$O(q_\theta(\mathbf{v}_{(h,r,t)})) = -\mathrm{KL}(q_\theta(\mathbf{v}_H)||p_w(\mathbf{v}_H|\mathbf{v}_O)) = \sum_{\mathbf{v}_H} q_\theta(\mathbf{v}_H)[\log p_w(\mathbf{v}_H|\mathbf{v}_O) - \log q_\theta(\mathbf{v}_H)]$$

$$= \sum_{\mathbf{v}_H} \prod_{(h',r',t')\in H} q_\theta(\mathbf{v}_{(h',r',t')}) \left[ \log p_w(\mathbf{v}_H,\mathbf{v}_O) - \sum_{(h',r',t')\in H} \log q_\theta(\mathbf{v}_{(h',r',t')}) \right] + \mathrm{const}$$

$$= \sum_{\mathbf{v}_{(h,r,t)}} \sum_{\mathbf{v}_{H\backslash(h,r,t)}} q_\theta(\mathbf{v}_{(h,r,t)}) \prod_{(h',r',t')\neq(h,r,t)} q_\theta(\mathbf{v}_{(h',r',t')}) \left[ \log p_w(\mathbf{v}_H,\mathbf{v}_O) - \sum_{(h',r',t')\in H} \log q_\theta(\mathbf{v}_{(h',r',t')}) \right] + \mathrm{const}$$

$$= \sum_{\mathbf{v}_{(h,r,t)}} q_\theta(\mathbf{v}_{(h,r,t)}) \sum_{\mathbf{v}_{H\backslash(h,r,t)}} \prod_{(h',r',t')\neq(h,r,t)} q_\theta(\mathbf{v}_{(h',r',t')}) \log p_w(\mathbf{v}_H,\mathbf{v}_O) -$$

$$\sum_{\mathbf{v}_{(h,r,t)}} q_\theta(\mathbf{v}_{(h,r,t)}) \sum_{\mathbf{v}_{H\backslash(h,r,t)}} \prod_{(h',r',t')\neq(h,r,t)} q_\theta(\mathbf{v}_{(h',r',t')}) \left[ \sum_{(h',r',t')\neq(h,r,t)} \log q_\theta(\mathbf{v}_{(h',r',t')}) + \log q_\theta(\mathbf{v}_{(h,r,t)}) \right] + \mathrm{const}$$

$$= \sum_{\mathbf{v}_{(h,r,t)}} q_\theta(\mathbf{v}_{(h,r,t)}) \log F(\mathbf{v}_{(h,r,t)}) - \sum_{\mathbf{v}_{(h,r,t)}} q_\theta(\mathbf{v}_{(h,r,t)}) \log q_\theta(\mathbf{v}_{(h,r,t)}) + \mathrm{const}$$

$$= -\mathrm{KL}\left( q_\theta(\mathbf{v}_{(h,r,t)})||\frac{F(\mathbf{v}_{(h,r,t)})}{Z} \right) + \mathrm{const}.$$
$$\tag{2}$$

Here, $Z$ is a normalization term, which makes $F(\mathbf{v}_{(h,r,t)})$ a valid distribution on $\mathbf{v}_{(h,r,t)}$, and we have

$$\log F(\mathbf{v}_{(h,r,t)}) = \sum_{\mathbf{v}_{H\backslash(h,r,t)}} \prod_{(h',r',t')\neq(h,r,t)} q_\theta(\mathbf{v}_{(h',r',t')}) \log p_w(\mathbf{v}_H,\mathbf{v}_O) = \mathbb{E}_{q_\theta(\mathbf{v}_{H\backslash(h,r,t)})}[\log p_w(\mathbf{v}_H,\mathbf{v}_O)].$$

Based on the Eq.(2), the optimal $q_\theta(\mathbf{v}_{(h,r,t)})$ is achieved when it equals to $\frac{F(\mathbf{v}_{(h,r,t)})}{Z}$, and thus we have:

$$\log q_\theta(\mathbf{v}_{(h,r,t)}) = \log F(\mathbf{v}_{(h,r,t)}) + \mathrm{const} = \mathbb{E}_{q_\theta(\mathbf{v}_{H\backslash(h,r,t)})}[\log p_w(\mathbf{v}_H,\mathbf{v}_O)] + \mathrm{const}$$

$$= \mathbb{E}_{q_\theta(\mathbf{v}_{H\backslash(h,r,t)})}\left[\log p_w(\mathbf{v}_{(h,r,t)}|\mathbf{v}_{O\cup H\backslash(h,r,t)})\right] + \mathrm{const}$$

$$= \mathbb{E}_{q_\theta(\mathbf{v}_{H\backslash(h,r,t)})}\left[\log p_w(\mathbf{v}_{(h,r,t)}|\mathbf{v}_{\mathrm{MB}(h,r,t)})\right] + \mathrm{const}$$

$$= \mathbb{E}_{q_\theta(\mathbf{v}_{\mathrm{MB}(h,r,t)\cap H})}\left[\log p_w(\mathbf{v}_{(h,r,t)}|\mathbf{v}_{\mathrm{MB}(h,r,t)})\right] + \mathrm{const}.$$

Here, $p_w(\mathbf{v}_{(h,r,t)}|\mathbf{v}_{O\cup H\backslash(h,r,t)}) = p_\phi(\mathbf{v}_{(h,r,t)}|\mathbf{v}_{\mathrm{MB}(h,r,t)})$ is derived from the conditional independence property of Markov networks.

## 1.2 Derivative w.r.t. $w$ in the M Step

In pLogicNet, we optimize $w$ through the gradient descent algorithm. For each expected conditional distribution $\mathbb{E}_{q_\theta(\mathbf{v}_H)}[p(\mathbf{v}_{(h,r,t)}|\mathbf{v}_{\mathrm{MB}(h,r,t)})]$, suppose that $\mathbf{v}_{(h,r,t)}$ connects with $\mathbf{v}_{\mathrm{MB}(ijk)}$ through a

18    set of rules. Then for each of such rules $l$, the derivative with respect to $w_l$ is computed as:

$$\nabla_{w_l}\mathbb{E}_{q_\theta(\mathbf{v}_H)}[\log p(\mathbf{v}_{(h,r,t)}|\mathbf{v}_{\text{MB}(h,r,t)})] \simeq \begin{cases} 1 - p_w(\mathbf{v}_{(h,r,t)} = 1|\hat{\mathbf{v}}_{\text{MB}(h,r,t)}) & \text{if } (h,r,t) \in O, \\ q_\theta(\mathbf{v}_{(h,r,t)} = 1) - p_w(\mathbf{v}_{(h,r,t)} = 1|\hat{\mathbf{v}}_{\text{MB}(h,r,t)}) & \text{if } (h,r,t) \in H, \end{cases}$$

19    where $\hat{\mathbf{v}}_{\text{MB}(h,r,t)} = \{\hat{\mathbf{v}}_{(h',r',t')}\}_{(h',r',t')\in\text{MB}(h,r,t)}$ is a sample from $q_\theta$. For each $(h',r',t') \in$
20    $\text{MB}(h,r,t)$, $\hat{\mathbf{v}}_{(h',r',t')} = 1$ if $(h',r',t')$ is observed, and otherwise $\hat{\mathbf{v}}_{(h',r',t')} \sim q_\theta(\mathbf{v}_{(h',r',t')})$.

21    Next, we prove the above claim. Specifically, we consider a logic rule $l$ connecting $\mathbf{v}_{(h,r,t)}$ with
22    $\mathbf{v}_{\text{MB}(h,r,t)}$, then we discuss two cases, i.e., $(h,r,t) \in O$ and $(h,r,t) \in H$.

23    **Case 1: $(h,r,t)$ is an observed triplet.**

24    Since $(h,r,t) \in O$, we have $\mathbb{E}_{q_\theta(\mathbf{v}_H)}[\log p(\mathbf{v}_{(h,r,t)}|\mathbf{v}_{\text{MB}((h,r,t))})] \simeq \log p(\mathbf{v}_{(h,r,t)} = 1|\hat{\mathbf{v}}_{\text{MB}((h,r,t))})$,
25    where $\hat{\mathbf{v}}_{\text{MB}(h,r,t)}$ is defined in the above paragraphs, and $p(\mathbf{v}_{(h,r,t)} = 1|\hat{\mathbf{v}}_{\text{MB}(h,r,t)})$ is computed as:

$$\begin{aligned} p(\mathbf{v}_{(h,r,t)} = 1|\hat{\mathbf{v}}_{\text{MB}(h,r,t)}) &= \frac{p(\mathbf{v}_{(h,r,t)} = 1, \hat{\mathbf{v}}_{\text{MB}(h,r,t)})}{p(\mathbf{v}_{(h,r,t)} = 1, \hat{\mathbf{v}}_{\text{MB}(h,r,t)}) + p(\mathbf{v}_{(h,r,t)} = 0, \hat{\mathbf{v}}_{\text{MB}(h,r,t)})} \\ &= \frac{\exp(w_l + b)}{\exp(w_l + b) + 1} = \sigma(w_l + b). \end{aligned} \tag{3}$$

26    Here, $b$ is a term which does not depend on $w_l$, and $\sigma$ is the sigmoid function, i.e., $\sigma(x) = \frac{1}{1+\exp(-x)}$.

27    Based on that, the derivative with respect to $w_l$ is computed as:

$$\begin{aligned} \nabla_{w_l}\mathbb{E}_{q_\theta(\mathbf{v}_H)}&[\log p(\mathbf{v}_{(h,r,t)}|\mathbf{v}_{\text{MB}(h,r,t)})] \simeq \nabla_{w_l}\log p(\mathbf{v}_{(h,r,t)} = 1|\hat{\mathbf{v}}_{\text{MB}(h,r,t)}) = \nabla_{w_l}\log\sigma(w_l + b) \\ &= \frac{\nabla_{w_l}\sigma(w_l + b)}{\sigma(w_l + b)} = \frac{\sigma(w_l + b)(1 - \sigma(w_l + b))}{\sigma(w_l + b)} = 1 - \sigma(w_l + b) = 1 - p(\mathbf{v}_{(h,r,t)} = 1|\hat{\mathbf{v}}_{\text{MB}(h,r,t)}). \end{aligned} \tag{4}$$

28    **Case 2: $(h,r,t)$ is a hidden triplet.**

29    Since $(h,r,t) \in H$, we have $\mathbb{E}_{q_\theta(\mathbf{v}_H)}[\log p(\mathbf{v}_{(h,r,t)}|\mathbf{v}_{\text{MB}(h,r,t)})] \simeq$
30    $\mathbb{E}_{q_\theta(\mathbf{v}_{(h,r,t)})}[\log p(\mathbf{v}_{(h,r,t)}|\hat{\mathbf{v}}_{\text{MB}(h,r,t)})]$, where $\hat{\mathbf{v}}_{\text{MB}(h,r,t)}$ is defined in the above paragraphs,
31    and $p(\mathbf{v}_{(h,r,t)} = 1|\hat{\mathbf{v}}_{\text{MB}(h,r,t)})$ is computed as:

$$\begin{aligned} p(\mathbf{v}_{(h,r,t)} = 1|\hat{\mathbf{v}}_{\text{MB}(h,r,t)}) &= \frac{p(\mathbf{v}_{(h,r,t)} = 1, \hat{\mathbf{v}}_{\text{MB}(h,r,t)})}{p(\mathbf{v}_{(h,r,t)} = 1, \hat{\mathbf{v}}_{\text{MB}(h,r,t)}) + p(\mathbf{v}_{(h,r,t)} = 0, \hat{\mathbf{v}}_{\text{MB}(h,r,t)})} \\ &= \frac{\exp(w_l + b)}{\exp(w_l + b) + 1} = \sigma(w_l + b). \end{aligned} \tag{5}$$

32    Here, $b$ is a term which does not depend on $w_l$, and $\sigma$ is the sigmoid function, i.e., $\sigma(x) = \frac{1}{1+\exp(-x)}$.

33    Based on that, the derivative with respect to $w_l$ is computed as:

$$\begin{aligned} \nabla_{w_l}\mathbb{E}_{q_\theta(\mathbf{v}_H)}&[\log p(\mathbf{v}_{(h,r,t)}|\mathbf{v}_{\text{MB}(h,r,t)})] \simeq \nabla_{w_l}\mathbb{E}_{q_\theta(\mathbf{v}_{(h,r,t)})}[\log p(\mathbf{v}_{(h,r,t)}|\hat{\mathbf{v}}_{\text{MB}(h,r,t)})] \\ &= \nabla_{w_l}[q_\theta(\mathbf{v}_{(h,r,t)} = 1)\log p(\mathbf{v}_{(h,r,t)} = 1|\hat{\mathbf{v}}_{\text{MB}(h,r,t)}) + q_\theta(\mathbf{v}_{(h,r,t)} = 0)\log p(\mathbf{v}_{(h,r,t)} = 0|\hat{\mathbf{v}}_{\text{MB}(h,r,t)})] \\ &= \nabla_{w_l}[q_\theta(\mathbf{v}_{(h,r,t)} = 1)\log\sigma(w_l + b) + q_\theta(\mathbf{v}_{(h,r,t)} = 0)\log(1 - \sigma(w_l + b))] \\ &= q_\theta(\mathbf{v}_{(h,r,t)} = 1)\frac{\sigma(w_l + b)(1 - \sigma(w_l + b))}{\sigma(w_l + b)} - q_\theta(\mathbf{v}_{(h,r,t)} = 0)\frac{\sigma(w_l + b)(1 - \sigma(w_l + b))}{1 - \sigma(w_l + b)} \\ &= q_\theta(\mathbf{v}_{(h,r,t)} = 1) - q_\theta(\mathbf{v}_{(h,r,t)} = 1)\sigma(w_l + b) - q_\theta(\mathbf{v}_{(h,r,t)} = 0)\sigma(w_l + b) = q_\theta(\mathbf{v}_{(h,r,t)} = 1) - \sigma(w_l + b) \\ &= q_\theta(\mathbf{v}_{(h,r,t)} = 1) - p_w(\mathbf{v}_{(h,r,t)} = 1|\hat{\mathbf{v}}_{\text{MB}(h,r,t)}). \end{aligned} \tag{6}$$

34    ## 1.3   Statistics of the Datasets

35    The statistics of the four datasets are presented in Tab. 1.

Table 1: Dataset statistics.

| Dataset | # Entities | # Relations | # Training | # Validation | # Test |
|---------|-----------|-------------|------------|--------------|--------|
| FB15k | 14,951 | 1,345 | 483,142 | 50,000 | 59,071 |
| WN18 | 40,943 | 18 | 141,442 | 5,000 | 5,000 |
| FB15k-237 | 14,541 | 237 | 272,115 | 17,535 | 20,466 |
| WN18RR | 40,943 | 11 | 86,835 | 3,034 | 3,134 |

Table 2: The best hyperparameter setting of pLogicNet on several benchmarks.

| Dataset | Embedding Dimension | Batch Size | # Negative Samples | $\alpha$ | $\gamma$ | Learning Rate |
|---------|---------------------|------------|---------------------|----------|----------|---------------|
| FB15k | 1000 | 2048 | 128 | 1.0 | 24 | 0.0001 |
| WN18 | 500 | 512 | 1024 | 0.5 | 12 | 0.0001 |
| FB15k-237 | 1000 | 1024 | 256 | 1.0 | 9 | 0.00005 |
| WN18RR | 500 | 512 | 1024 | 0.5 | 6 | 0.00005 |

## 1.4 Hyperparameters

In pLogicNet, we parameterize the variational distribution $q_\theta$ as a TransE model [1], and we use the method as used in [3] for training the model. More specifically, we define $q_\theta(\mathbf{v}_{(h,r,t)})$ by using a distance-based formulation, i.e., $q_\theta(\mathbf{v}_{(h,r,t)} = 1) = \sigma(\gamma - ||\mathbf{x}_h + \mathbf{x}_r - \mathbf{x}_t||)$, where $\sigma$ is the sigmoid function and $\gamma$ is a hyperparameter, which is fixed during training. We generate negative samples by using self-adversarial negative sampling [3], and use Adam [2] as the optimizer. The detailed parameter settings can be found in Tab. 2.