[Reviews · NeurIPS 2019]

Reviewer 1



This paper solves the task of knowledge base completion i.e. filling the missing relations between two entities by combining Statistical Relational Model like Markov Logic, and knowledge graph embedding method like TransE. Authors define a set of rules to be used in MLNs and then define a joint probability distribution over the observed and hidden triplets. Similarly, they define a joint probability distribution using KGE approaches (specifically they chose transE model). Then they employ the variational EM algorithm to learn the MLN weights and finally predicting the probabilities of hidden triplets. Originality: I really liked the paper, and enjoyed thoroughly reading it. Although people have looked into this idea of combining rule-based and KGE approaches in several papers, the idea of combining a pure SRL model like Markov Logic is new AFAIK. Quality: The paper has an adequate amount of theory, complemented well with the experiments on well-known 4 datasets. The ablation study, both in terms of KGE models and rules of the MLN shows the effectiveness of their approach. However, I have the following issues: (1) they haven't provided details about the rules of MLNs they have created. In the code also, I couldn't find the details about the exact MLN rules they created. For example, for what relations did they create formulas of MLNs?, (2) They have learned MLN weights using a vanilla gradient descent, however almost all the current approaches using MLN use a much better method like prescaled conjugate gradient descent or LBFGS in case of pseudologlikelihood. Do you have a specific reason to not choose those methods? (3) Moreover, they have yet to compare their method against the SOTA method like RotatE. Clarity: The paper is written very clearly, and necessary theorems and proofs have been provided. Significance: This is a significant work, but the lack of comparison against the SOTA method makes it a weak accept. Some typos: Line 143: Varational -> Variational Eq 6: should be p_w instead of p UPDATE: I have read the author's feedback and am convinced with their response. I vote for accepting this work for the poster presentation.

Reviewer 2



This paper shows introduces a way to combine a markov (logic) network with knowledge graph embeddings. In particular, the approach uses EM to train the weights of a Markov Logic Network in the M-step while inferring latent triple states using a KG embedding model as variational distribution in the E-step. Results on various standard benchmarks are convincing. I think this paper is well written and relatively clear. The idea is straightforward but in a good way (the kind of thing I thought people would have tried much earlier but haven't). The results are convincing and reasonably ablated. There are various approximations/heuristics that the authors use to make this tractable (e.g. sampling the markov blanket before calculating the expectation). These have fewer theoretical groundings but the empirical results justify them. The paper could do a better job in discussing recent related work such as "Learning Explanatory Rules from Noisy Data", "End-to-End Differentiable Proving" and "Adversarial Sets for Regularising Neural Link Predictors" that are either related or very related (the last one for example).

Reviewer 3



The paper is clearly written. I find the variational EM training of MLNs to be interesting. In the E-step, it cleverly uses graph embeddings to sample the truth values of hidden triplets, thereby completely defining the Markov blanket of each triplet for the M-step. This circumvents the need for (intractable) inference to obtain the values of the hidden triplets. However, I do not find the M-step to be novel because the maximization of pseudolikelihood for MLNs is standard fare (e.g., [20]). I have two misgivings about the paper, both of which relate to recent hybrid approaches RUGE [15] and NNE-AER [9]. First, the paper does not position its proposed system clearly vis-a-vis RUGE and NNE-AER. Both systems combine knowledge graph embeddings with first-order logic, and their logical rules are soft and hence capture their inherent uncertainty. The rules are also encapsulated in a maximization equation in a principled manner. Hence, both systems seem to integrate soft logical rules with graph embeddings in as principled a manner as the paper's system. This begs the question of why the paper's system does better than RUGE and NNE-AER in the experiments. Second, I do not think the experimental comparison with RUGE and NNE-AER are truly apples-to-apples. The empirical numbers are taken from the RUGE and NNE-AER papers. In those papers, NNE-AER only considers "non-negativity" rules and "approximate entailment" rules, and RUGE only considers Horn clauses of length at most 2. In contrast, this paper considers four rule types: composition rules, inverse rules, symmetric rules, and subrelation rules. Hence, it is possible that the paper is doing better than RUGE and NNE-AER simply because it considers a different (and possibly bigger) universe of rules, rather than through a more principled combination of soft logic and graph embeddings. UPDATE: In their feedback, the authors have sufficiently addressed my two misgivings with regards to RUGE and NNE-AER. Hence, I will upgrade my score to "marginally above acceptance".

[Author Response · NeurIPS 2019]

First of all, we would like to thank all the anonymous reviewers for the insightful comments and suggestions!

**To Reviewer 1:**

**(1) Extracted rules.** The number of first-order formulas is 29363, 64, 919, 19 on FB15k, WN18, FB15k-237, WN18RR respectively. We show the top-ranked rules and the weights learned by our approach on FB15k in the table below.

| Composition | | Symmetric | |
|---|---|---|---|
| [Person] *Place of Birth* [Location] ∧ [Location] *In Country* [Country] ⇒ [Person] *Nationality* [Country] | 4.06 | [Location] *Adjoins* [Location] | 3.52 |
| [Film] *Film Actor* [Actor] ∧ [Actor] *Dubbing Language* [Language] ⇒ [Film] *Film Language* [Language] | 2.41 | [Person] *Spouse* [Person] | 3.46 |

| Inverse | | Subrelation | |
|---|---|---|---|
| [Person] *Played* [Instrument] ⇒ [Instrument] *Instrumentalists* [Person] | 8.74 | [Award] *Winner* [Person] ⇒ [Award] *Nominee* [Person] | 3.04 |
| [Location] *In Time Zone* [Time Zone] ⇒ [Time Zone] *Time Zone of* [Location] | 5.17 | [Person] *Honored* [Award] ⇒ [Person] *Nominated* [Award] | 3.02 |

**(2) Learning algorithms for MLNs.** For learning MLNs, we tried L-BFGS as you suggested, and we got similar results as gradient descent. The reason is that learning MLNs with pseudolikelihood is a convex optimization problem, and therefore both gradient descent and L-BFGS are able to converge to the global optima, although L-BFGS is faster.

**(3) Comparison with RotatE.** The comparison with RotatE on the two larger datasets (i.e., FB15k and WN18) is shown in the right table, where we use RotatE as our KGE

| Algorithm | FB15k | | | | | WN18 | | | | |
|---|---|---|---|---|---|---|---|---|---|---|
| | MR | MRR | H@1 | H@3 | H@10 | MR | MRR | H@1 | H@3 | H@10 |
| RotatE | **40** | 0.797 | 74.6 | 83.0 | 88.4 | 309 | 0.949 | 94.4 | 95.2 | 95.9 |
| pLogicNet | 42 | **0.815** | **77.6** | **83.8** | **88.7** | 256 | **0.950** | **94.5** | **95.3** | **96.1** |

model. The improvement of pLogicNet over RotatE is not as significant as over TransE, as RotatE can already implicitly model most important rules on the current datasets. However, with MLNs, our framework is general to incorporate any logic rules, many of which could not be modeled by RotatE. This could be quite advantageous in some domains.

**To Reviewer 2:**

**(1) Related work.** Thanks for pointing out these papers! The first two papers focus on inductive logic programming combined with neural networks, whereas our paper focuses on statistical relational reasoning (Markov Logic Networks in specific) combined with knowledge graph embedding for knowledge graph reasoning. The third paper is the most relevant one to ours, which uses Horn clauses to regularize the predictions made by knowledge graph embeddings. In contrast, our approach is built on top of the principled probabilistic framework, Markov Logic Networks, which can effectively handle the uncertainty of logic rules and triplets. We will discuss these papers in details in the revised draft.

**(2) MLN settings.** Thanks for the suggestions! Our approach is indeed quite general and can be applied to those MLN settings. We will add the results under the MLN settings in the future.

**To Reviewer 3:**

**(1) Conceptual comparison.** Thanks for the correction!! Both RUGE and NNE-AER are elegant methods for combining logic rules and knowledge graph embedding. As you corrected, both methods can also model the uncertainty of logic rules by using soft rules. However, our method has some key differences from RUGE and NNE-AER.

First, our approach models the uncertainty of logic rules under the principled framework of Markov Logic Networks. In all the three methods, each rule is associated with a weight to measure the uncertainty. In both RUGE and NNE-AER, the weights of rules are initialized by other algorithms (i.e., AMIE+) and then fixed during training. In contrast, our approach can dynamically learn and adjust the weights of logic rules according to the downstream reasoning tasks.

Second, both RUGE and NNE-AER are specifically designed for the task of knowledge graph reasoning. In contrast, as both Reviewer 1 and 2 pointed out, our approach is a general mechanism. Besides knowledge graph reasoning, our approach can also be applied to many other tasks in statistical relational reasoning. Therefore, our approach could be valuable to the statistical relational learning community, and no existing approaches have studied combining statistical relational learning methods (Markov Logic Networks in particular) with knowledge graph embedding methods.

**(2) Apples-to-apples comparison.** Thanks for pointing this out!! We have done some experiments under the same settings as used in RUGE and NNE-AER. Following both methods, we use ComplEx as the KGE model. To compare with RUGE, we only use Horn clauses of length at most 2, corresponding to our inverse and composition rules (see Table 1 of the RUGE paper). To compare with NNE-AER, we only use the approximate entailment, corresponding to our inverse and subrelation rules (see Table 1 of the NNE-AER paper). We present the results in the following table, where our approach consistently performs better. The reason is that our approach can dynamically infer the uncertainty of logic rules based on the downstream reasoning tasks. For the results of our approach on FB15k, one may notice that the results in the right table are higher than those in the left table. This is because in the right table, no composition rules are used. As the composition rules on the FB15k dataset are very noisy, incorporating them will decrease the results.

| Algorithm | FB15k | | | |
|---|---|---|---|---|
| | MRR | H@1 | H@3 | H@10 |
| RUGE | 0.768 | 70.3 | 81.5 | 86.5 |
| pLogicNet | **0.786** | **73.3** | **82.0** | **88.2** |

| Algorithm | FB15k | | | | WN18 | | | |
|---|---|---|---|---|---|---|---|---|
| | MRR | H@1 | H@3 | H@10 | MRR | H@1 | H@3 | H@10 |
| NNE-AER | 0.803 | 76.1 | 83.1 | 87.4 | 0.943 | 94.0 | 94.5 | 94.8 |
| pLogicNet | **0.827** | **79.2** | **84.6** | **89.3** | **0.950** | **94.3** | **95.6** | **96.3** |

[Meta-Review · NeurIPS 2019]

The reviewers felt the paper presents a significant bridge between logical modeling and knowledge graph embeddings. The author response presented some improved analysis of the experiments and context in comparing against existing approaches that should be incorporated into the final version.